# Feed-forward alpha particle radiotherapy ablates androgen receptor-addicted prostate cancer

Michael R. McDevitt [1,2], Daniel L.J. Thorek[3,4], Takeshi Hashimoto[5], Tatsuo Gondo[5], Darren R. Veach [1,2,6], Sai Kiran Sharma[6], Teja Muralidhar Kalidindi[1], Diane S. Abou[3], Philip A. Watson[7], Bradley J. Beattie[8], Oskar Vilhemsson Timmermand[9], Sven-Erik Strand [10], Jason S. Lewis [1,2,6,11], Peter T. Scardino[12,13], Howard I. Scher[13,14], Hans Lilja [12,14,15,16,17], Steven M. Larson[1,2,11,18] & David Ulmert[9,11]

Human kallikrein peptidase 2 (hK2) is a prostate specific enzyme whose expression is governed by the androgen receptor (AR). AR is the central oncogenic driver of prostate cancer (PCa) and is also a key regulator of DNA repair in cancer. We report an innovative therapeutic strategy that exploits the hormone-DNA repair circuit to enable molecularly-specific alpha particle irradiation of PCa. Alpha-particle irradiation of PCa is prompted by molecularly specific-targeting and internalization of the humanized monoclonal antibody hu11B6 targeting hK2 and further accelerated by inherent DNA-repair that up-regulate hK2 (KLK2) expression in vivo. hu11B6 demonstrates exquisite targeting specificity for KLK2. A single administration of actinium-225 labeled hu11B6 eradicates disease and significantly prolongs survival in animal models. DNA damage arising from alpha particle irradiation induces AR and subsequently KLK2, generating a unique feed-forward mechanism, which increases binding of hu11B6. Imaging data in nonhuman primates support the possibility of utilizing hu11B6 in man.

[1] Department of Radiology, Memorial Sloan Kettering Cancer Center, New York, NY 10065, USA. [2] Department of Radiology, Weill Cornell Medical College, New York, NY 10065, USA. [3] Division of Nuclear Medicine and Molecular Imaging, Department of Radiology and Radiological Science, Sidney Kimmel Comprehensive Cancer Center, Johns Hopkins School of Medicine, Baltimore, MD 21205, USA. [4] Cancer Molecular and Functional Imaging Program, Department of Oncology, Sidney Kimmel Comprehensive Cancer Center, Johns Hopkins School of Medicine, Baltimore, MD 21205, USA. [5] Department of Urology, Tokyo Medical University, 6-7-1 Nishi-shinjuku, Shinjuku-ku, Tokyo 160-0023, Japan. [6] Radiochemistry and Imaging Sciences Service, Department of Radiology, Memorial Sloan Kettering Cancer Center, New York, NY 10065, USA. [7] Human Oncology and Pathogenesis Program, Memorial Sloan Kettering Cancer Center, New York 10065 NY, USA. [8] Department of Medical Physics, Memorial Sloan Kettering Cancer Center, New York, NY 10065, USA. [9] Division of Oncology, Clinical Sciences, Lund University and Skåne University Hospital, Barngatan 4, 22100 Lund, Sweden. [10] Department of Clinical Sciences, Medical Radiation Physics, Lund University, Barngatan 4, 22100 Lund, Sweden. [11] Molecular Pharmacology Program, Sloan Kettering Institute, Memorial Sloan Kettering Cancer Center, New York, NY 10065, USA. [12] Urology Service, Department of Surgery, Memorial Sloan Kettering Cancer Center, New York, NY 10065, USA. [13] Department of Urology, Weill Cornell Medical College, New York, NY 10065, USA. [14] Genitourinary Oncology Service, Department of Medicine, Memorial Sloan Kettering Cancer Center, New York, NY 10065, USA. [15] Department of Laboratory Medicine, Memorial Sloan Kettering Cancer Center, New York, NY 10065, USA. [16] Nuffield Department of Surgical Sciences, University of Oxford, Oxford OX3 7DQ, UK. [17] Department of Translational Medicine, Lund University, J Waldenströms gata 35, 20502 Malmö, Sweden. [18] Nuclear Medicine Service, Department of Radiology, Memorial Sloan Kettering Cancer Center, New York, NY 10065, USA. These authors contributed equally: Michael R. McDevitt, Daniel L. J. Thorek.  Correspondence and requests for materials should be addressed to D.U. (email: ulmerth@mskcc.org)

Alpha particles (α-particles) are potent therapeutic effectors that have entered clinical practice[1–3]. These charged helium nuclei emitted upon decay travel approximately 50–80 μm and have a high linear energy transfer (LET) of approximately 100 keV/μm with high relative biological effect. Thus, they are able to kill a target cell by depositing 5–8 MeV in a highly focused ionizing track that is only several cell diameters in length. Indeed, a single α-particle traversal through a cell can be cytotoxic[4, 5]. Unlike the approved bone seeking calcium mimetic, Radium-223 dichloride (Xofigo, Bayer Healthcare) Actinium-225 ($^{225}$Ac ($t_{1/2} = 10$ days)) is an α-particle emitting radionuclide that can be conjugated to targeting macromolecules such as antibodies for cancer cell specific therapy. We have deployed molecular targeting in our pharmacologic strategy to treat prostate cancer by exploiting upregulation of the androgen receptor (AR), a central oncogenic driver of the disease.

The overexpression of the AR, even in states of castrate resistant disease, is a hallmark of prostate cancer progression[6, 7]. Recent insights into the mechanisms of DNA repair in prostate cancer cells have shown that AR is associated with a plethora of DNA repair genes and also that active AR-signaling increases repair rates[8, 9]. The genotoxic consequences of α-particle irradiation to cancer cells result in significant, often lethal DNA damage.

In this study, we introduce a strategy that takes advantage of the oncoaddiction of prostate cancer to AR—and its upregulation following DNA damage—by radioimmunotherapy targeting of human kallikrein peptidase 2 (hK2). hK2 is a well-characterized protease, with 80% amino acid (a.a.) identity to prostate specific antigen (PSA), that is directly regulated by AR activity. A monoclonal antibody (hu11B6) has been designed to specifically address an epitope accessible only on the free, catalytically active form of human hK2 and has been developed as a diagnostic[10–12]. Importantly, when hu11B6 binds to active hK2, the immune complex is internalized by the cell and trafficked to lysosomal compartments in a process made possible by concomitant binding of the hu11B6 with neonatal Fc receptor (FcRn). Here we introduce a molecularly specific alpha particle emitting radiotherapeutic [$^{225}$Ac]hu11B6. Internalization of $^{225}$Ac-radiolabeled hu11B6 drug by the prostate cancer cell increases the probability that the α-particle energy is deposited in the nucleus.

Our overall strategy seeks to take full advantage of the unique consequences of [$^{225}$Ac]hu11B6-tissue-specific targeting and high LET α-particle irradiation of prostate cancer. We recognized that the α-particle induced DNA damage and ensuing upregulation of AR and *KLK2* would increase the prostate cancer targeting by [$^{225}$Ac]hu11B6, creating an amplification loop for cell-specific therapy. Using advanced small animal models of prostate cancer, we demonstrate the ability to effect disease control. Additionally, we establish a diagnostic imaging reporter for [$^{225}$Ac]hu11B6 evaluated in both murine models and nonhuman primates.

## Results

**Radiochemistry.** The radiochemical yield of [$^{225}$Ac]hu11B6 was 3.7% ± 2.1% (mean ± standard deviation, $n = 13$) and the radiochemical purity was 99.3% ± 0.5% ($n = 13$); specific activity was 0.079 Ci/g ± 0.055 Ci/g ($n = 13$). [$^{89}$Zr]hu11B6 was obtained in 93.0% radiochemical yield and was >99% pure ($n = 5$). The specific activity ranged from 0.88 to 1.1 Ci/g ($n = 5$).

**Binding affinity was not affected by radiolabeling.** The unmodified 11B6 antibody and its derivative radioimmunoconjugates demonstrated robust binding and dissociation curves with the purified hK2 on the protein A sensor chip. All the constructs that were analyzed yielded $K_D$ values in the lower nM

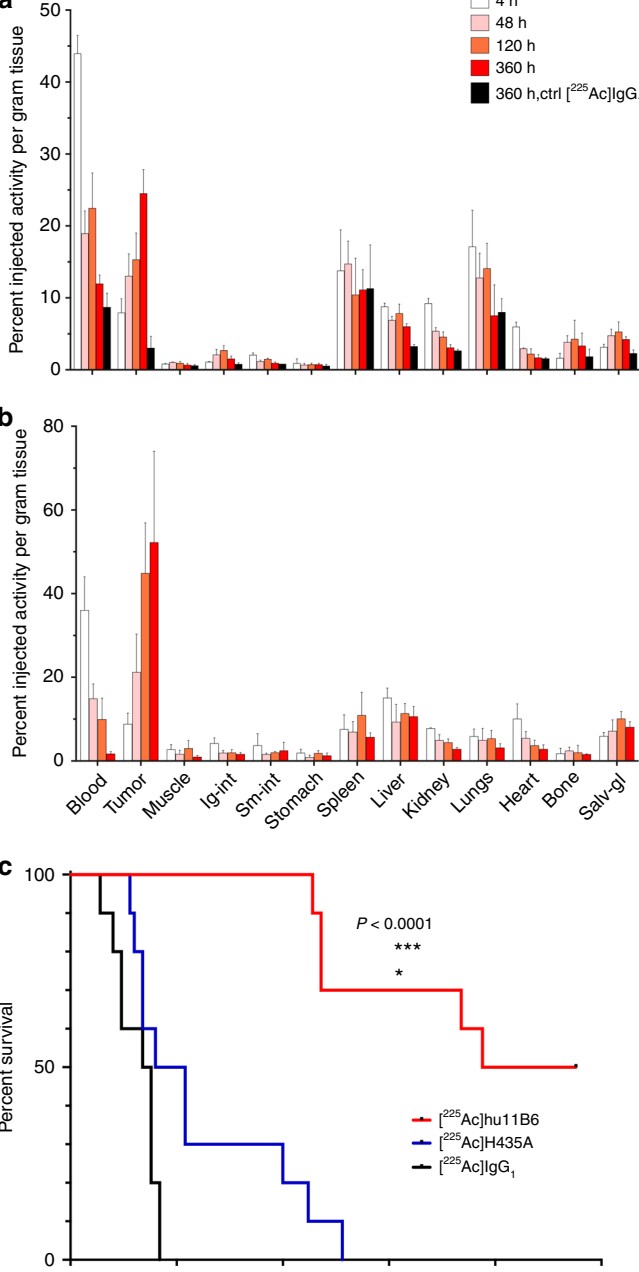

**Fig. 1** Pharmacokinetics and pharmacodynamics of [$^{225}$Ac]hu11B6 in xenograft models of prostate cancer. Full [$^{225}$Ac]hu11B6 biodistribution data set of (**a**) LNCaP-AR and (**b**) VCaP s.c. xenograft models, where the significantly higher tumor accumulation of [$^{225}$Ac]hu11B6 can be noted in the VCaP model as a result of higher target expression. **c** Kaplan–Meier plot comparing survival of LNCaP-AR s.c. tumor models ($n = 10$ per group) treated with a single 300 nCi dose of [$^{225}$Ac]hu11B6 (red line), 300 nCi non-internalizing hK2 targeting antibody [$^{225}$Ac]hu11B6-H435A (blue line), or 300 nCi of an alpha-particle labeled non-specific antibody [$^{225}$Ac]huIgG$_1$ (black line). These results demonstrate that effective hK2-targeted alpha particle delivery by hu11B6 depends on both antigen specificity and FcRn-mediated internalization of the immunoglobulin

range. The kinetic constants of [$^{225}$Ac]hu11B6 and [$^{89}$Zr]hu11B6 matched very well with those of the unmodified batch of 11B6 antibody that was used to prepare the radiolabeled constructs—thus indicating minimal loss of immunoreactivity of the

radioimmunoconjugates for binding to the target antigen. The 7.25 nM $K_D$ of the [89Zr]hu11B6 matched the $K_D$ of the native 11B6 antibody (5.48 nM) that was used to prepare the radio-immunoconjugate. Similarly, the 16.6 nM $K_D$ of the [225Ac]hu11B6 matched the 14 nM $K_D$ for the batch of native 11B6 antibody that was used to prepare this radioimmunoconjugate.

**Pharmacokinetics of [225Ac]hu11B6 in PCa xenografts.** Bio-distribution studies were conducted to evaluate the pharmaco-kinetic distribution of [225Ac]hu11B6 in human prostate cancer xenograft models (Fig. 1a,b). Animals with LNCaP-AR and VCaP tumors both exhibited [225Ac]hu11B6 persistence in the blood through 15 days (blood $t_{1/2}$ is approximately 2 days in both models). VCaP tumors have greater AR activity and higher hK2 expression than LNCaP-AR. Therefore, as predicted, VCaP accumulated more [225Ac]hu11B6 (mean %IA/g = 52.2 ± 21.8% at 15 days) compared to LNCaP-AR (mean %IA/g = 24.5 ± 3.3% at 15 days). Furthermore, the increased uptake in VCaP tumor permitted greater clearance from blood compared to LNCaP-AR disease (mean 1.7 ± 0.54 %IA/g vs. 12.0 ± 1.2 %IA/g, respectively at 15 days). The distribution of [225Ac]hu11B6 to other organs was minimal (approximately 11 %IA/g or less) and nonspecific as compared to an isotype matched human IgG control.

**Alpha irradiation of a PCa xenograft with [225Ac]hu11B6.** LNCaP-AR tumor bearing animals were treated with the (a) specific [225Ac]hu11B6 drug, (b) specific but non-internalizing [225Ac]hu11B6-H435A (this mutated antibody targets hK2 expressing cells but is inhibited from internalization via FcRn-coupled cellular processes by exchanging histidine 435 to an alanine), or (c) non-specific [225Ac]huIgG1 control antibody.

Kaplan–Meier analysis of these three treatments reports median survival of 108, 22.5, and 18 days from administration of 300 nCi of [225Ac]hu11B6, [225Ac]hu11B6-H435A, or [225Ac]huIgG1, respectively (Fig. 1c). At the conclusion of the study, half of the animals treated with the specific and internalizing [225Ac]hu11B6 drug were alive. These data show that [225Ac]hu11B6 treatment significantly enhanced the animals' survival compared to either control drug treatment ($P < 0.0001$) by Student's $t$-test. Efficacy is significantly dependent on a combination of specific targeting and cellular internalization of the alpha-emitting compound by the tumor.

**Microdistribution of [223Ra]RaCl2 and [225Ac]hu11B6.** Auto-radiograms of tissue from mice treated with [223Ra]RaCl2 or [225Ac]hu11B6 demonstrated distinctly different activity micro-distributions (Fig. 2). While [225Ac]hu11B6 localized directly to the tumor (indicated by black arrow in Fig. 2a), 223Ra showed accumulation to apposite bone surface surrounding the lesion, as well active bone modeling/remodeling sites. As visualized by subsequent tissue sections developed using Safranin-O, which differentiates mineral bone (green colored staining effect) from proteoglycans and cartilage (red-orange colored staining effect); 223Ra showed binding to the ossification front behind the growth plates (Fig. 2b).

**[225Ac]hu11B6 in a GEM model of human PCa.** A complete biodistribution study of [225Ac]hu11B6 was performed in a genetically modified mouse model (Hi-*Myc* x pb_KLK2) that has prostate specific expression of human hK2 (pb_KLK2) and spontaneous development of prostate adenocarcinoma (Hi-*Myc*) at 20–30 weeks of age (Fig. 3a). [225Ac]huIgG1 antibody was used

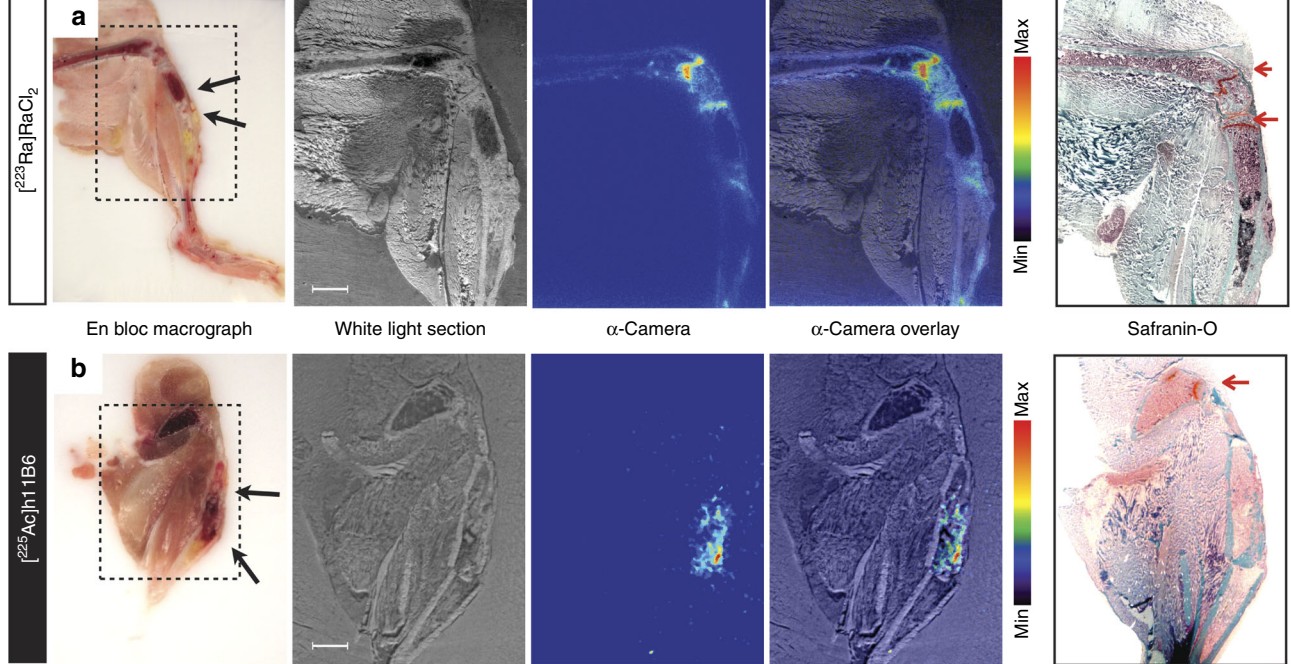

**Fig. 2** Comparative microdistribution of [223Ra]Cl2 and [225Ac]hu11B6 in mice bearing intratibial LNCaP-AR disease showed a differential uptake in vivo. Autoradiograms of (**a**) [223Ra]Cl2 and (**b**) [225Ac]hu11B6 demonstrated distinctly different microdistributions of activity. While [225Ac]hu11B6 localized directly to the tumor (indicated by black arrows in macrographs), [223Ra]Cl2 showed accumulation in apposite bone surface surrounding the lesion, as well active sites of both bone modeling and remodeling (n.b., disease is comprised of both osteolytic and osteoblastic bone lesions). Sequential tissue sections developed using Safranin-O, which differentiates mineral bone (green colored staining effect) from proteoglycans and cartilage (red-orange colored staining effect). The mineralizing front behind the growth plate at the epiphyses (red arrow) as shown in the Safranin-O stain show intense 223Ra uptake (Fig. 2a); antibody-targeted 225Ac does not accumulate at these sites (Fig. 2b)

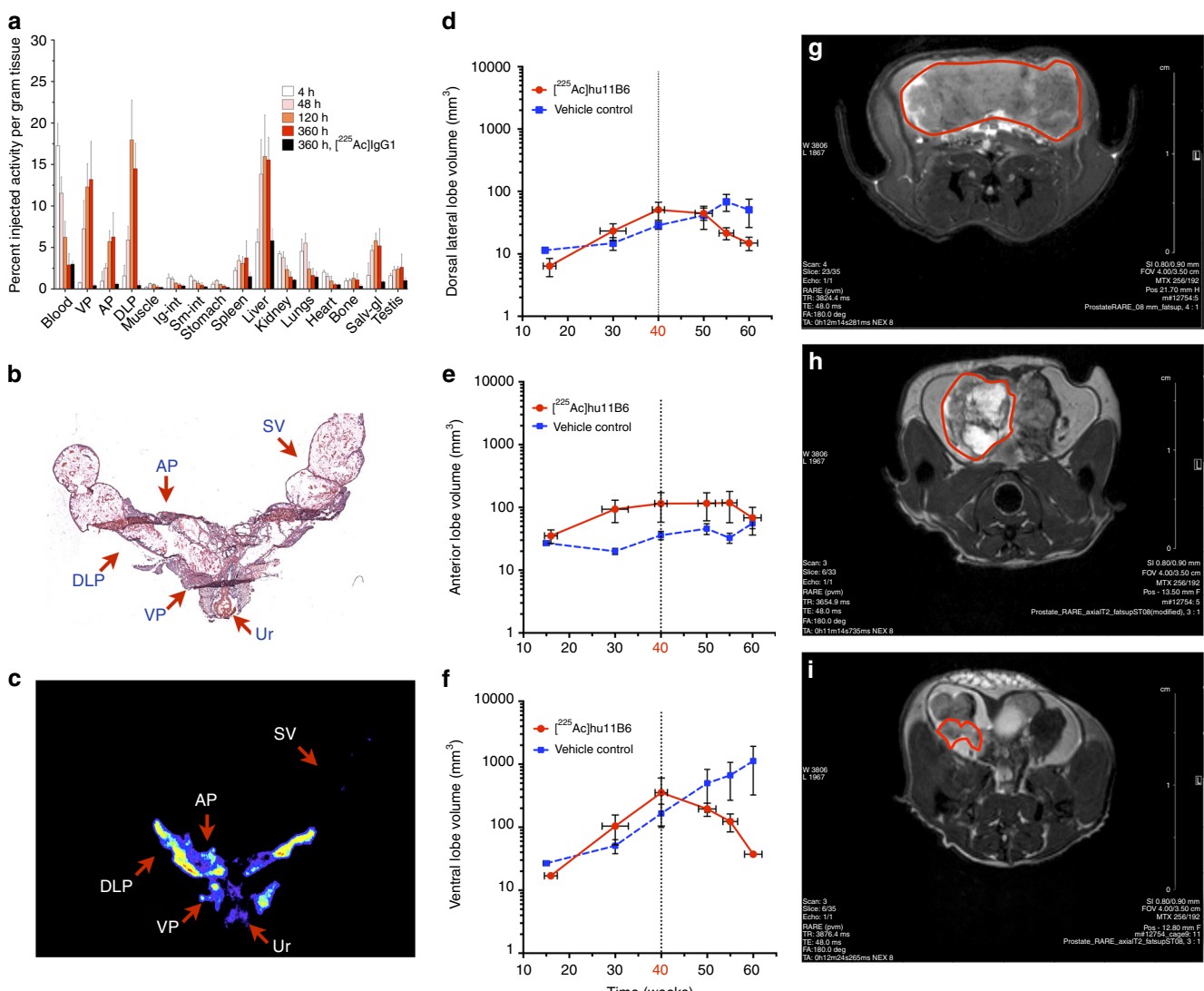

**Fig. 3** Specific and potent therapeutic effect of [$^{225}$Ac]hu11B6 in the Hi-*Myc* x pb_*KLK2* mouse model that expresses human hK2 in the prostate. **a** The pharmacokinetic profile of [$^{225}$Ac]hu11B6 was determined in a biodistribution experiment using genetically modified mice (Hi-*Myc* x pb_*KLK2*) with inherent prostate-specific expression of human hK2 and spontaneous development of prostate adenocarcinoma. This biodistribution data set shows increasing accumulation of drug in the ventral prostate (VP), dorsal-lateral prostate (DLP), and anterior prostate (AP) lobes with time (n.b., the VP and DLP exhibit higher expressions of hK2 and showed higher [$^{225}$Ac]hu11B6 uptake compared to AP tissue). H&E (**b**) and autoradiography (**c**) of a tissue section (encompassing all three prostate lobes, urethra and seminal vesicles) from a pb_*KLK2* x Hi-*Myc* injected with [$^{225}$Ac]hu11B6. Radioactivity is only associated with the prostate lobes and not with seminal vesicles (SV) and radiosensitive bystander urothelium (Ur). In a therapeutic study of [$^{225}$Ac] hu11B6, disease was monitored by volumetric MRI measurements of the individual lobes of Hi-*Myc* x pb_*KLK2* mice. Mice underwent longitudinal MRI commencing at 15–16 weeks of age and received a single 300 nCi dose of [$^{225}$Ac]hu11B6 at week 40. Lobe volumes are plotted vs. time following the effect of alpha irradiation on the DLP (**d**), AP (**e**), and VP (**f**) respectively. Vehicle-treated animals served as control for tumor progression. The results show that one injection of [$^{225}$Ac]hu11B6 significantly decreased tumor burden and inhibited disease progression in this GEM model of locally advanced disease. MR images of a representative animal confirms the significant reduction in tumor volume as shown at baseline (pre-treatment) (**g**) and at 4 (**h**) and 8 (**i**) weeks following a single injection of [$^{225}$Ac]hu11B6

as control to investigate [$^{225}$Ac]hu11B6 specificity at 360 h post-administration. [$^{225}$Ac]hu11B6 again displayed persistence in the blood through 15 days (blood $t_{1/2}$ is approximately 3 days in this model). The ventral prostate (VP) and dorsal-lateral prostate (DLP) lobes have higher hK2 expression and a greater incidence of epithelial dysplasia and carcinoma compared to the anterior prostate (AP) lobes, and as a consequence demonstrated a greater uptake of [$^{225}$Ac]hu11B6. At 15 days post-injection the VP, DLP, and AP had $13.3 \pm 4.6$, $14.5 \pm 3.0$, and $6.3 \pm 2.9$ %IA/g, respectively. In this model, liver showed the highest off-target accumulation which was approximately 16 %IA/g at 15 days. The

nontargeting [$^{225}$Ac]huIgG$_1$ had <1%IA/g accumulation in the VP, DLP, and AP but similar blood clearance and tissue accumulation compared to [$^{225}$Ac]hu11B6 at 15 days. The prostate lobes, urethra and seminal vesicles were removed from a representative animal in the biodistribution study and fixed and embedded for haematoxylin and eosin staining (Fig. 3b) and autoradiography (Fig. 3c). Radioactivity is associated only with the prostate lobes and absent in seminal vesicles and radiosensitive bystander tissues such as the urothelium. The specific localization of [$^{225}$Ac]hu11B6 to the targeted lobes of prostate with disease ensures that the absorbed dose from the short range

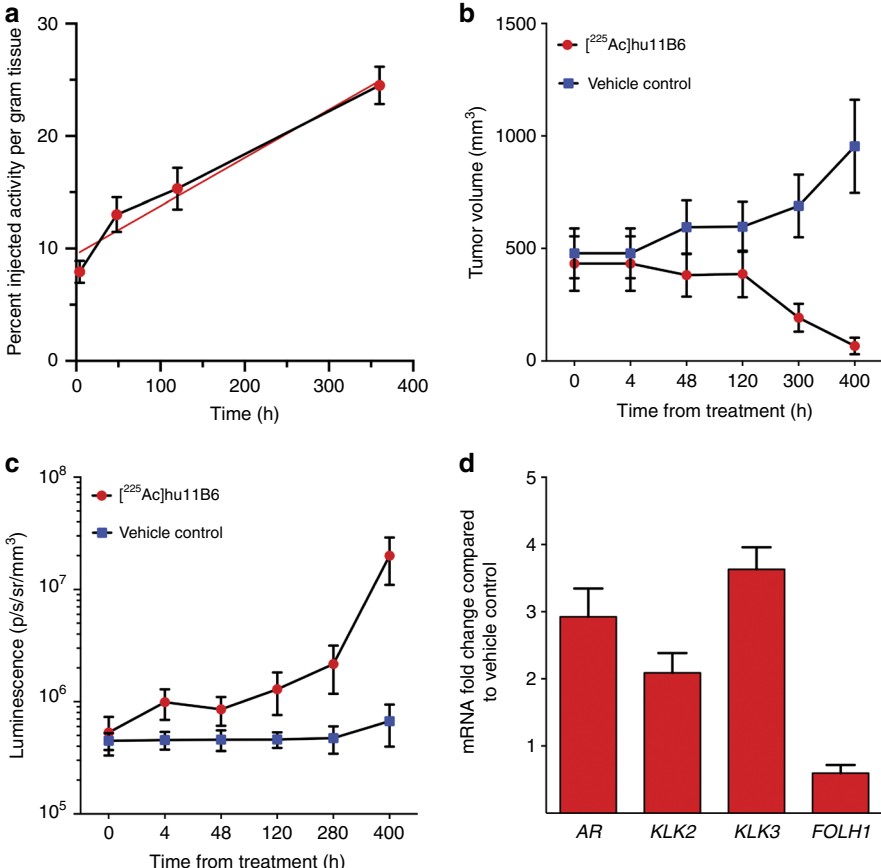

**Fig. 4** Alpha particles promote a feed-forward oncoaddictive effect that promulgates lethality. **a** The increasing uptake of [$^{225}$Ac]hu11B6 is shown as %IA/g (red-filled circles with a blue connecting line) in LNCaP-AR tumor at four time-points after IV administration. This escalating rate of drug accumulation in tumor is noted by the linear regression fit (solid red line) with slope = 0.037 ± 0.001 %IA/g/h. **b** Therapeutic efficacy of a single administration of [$^{225}$Ac] hu11B6 is evidenced by the decreasing tumor volumes compared with vehicle-treated controls where the trajectory of untreated tumor volumes steadily increase with time. **c** These data report the measured AR-driven luciferase bioluminescence signal (photons/s/mm$^2$/sr) vs. time in the treated and control animals. Note that while the alpha irradiated tumor volume decreases (see panel 4B) with time while AR-driven BLI signal increases. These data show upregulation of *AR* expression as a consequence of alpha irradiation with [$^{225}$Ac]hu11B6. **d** RT-PCR analysis of *AR* and *AR*-driven genes in tumor tissues collected from [$^{225}$Ac]hu11B6- and vehicle-treated mice at 400 h. The fold-change in tumor gene activity of treated animals normalized to non-treated tumors shows increased expression of *AR*, *KLK2*, and *KLK3*, but not *FOLH1*

alpha particles is deposited locally at sites of carcinoma, and not to healthy surrounding tissue.

**MRI quantification of [$^{225}$Ac]hu11B6 radiobiological effects.** 40 week old male pb_*KLK2* x Hi-*Myc* mice were randomized into two cohorts to receive (a) a single IV injection of 300 nCi of [$^{225}$Ac]hu11B6 or (b) vehicle. The individual prostate lobe volumes of all mice were measured by MRI at 15, 30, 40 (treatment start), 50, 55 and 60 weeks of age. Analysis of the imaging data as a function of time showed that [$^{225}$Ac]hu11B6 induced a large decrease in the DLP volume (14.9 ± 8.1 mm$^3$) compared to that without treatment (51.1 ± 60.0 mm$^3$) (Fig. 3d); a small decrease in AP volume (68.2 ± 71.5 mm$^3$) compared to that without treatment (55.7 ± 24.2 mm$^3$) (Fig. 3e), and a significant decrease in VP lobe volume (37.2 ± 6.0 mm$^3$) compared to that without treatment (1124 ± 1955 mm$^3$) (Fig. 3f) over a period of two weeks. The results show that one injection of [$^{225}$Ac] hu11B6 significantly decreased tumor burden and inhibits disease progression. MR imaging of the prostate of a representative animal from this study are shown before [$^{225}$Ac]hu11B6 treatment (Fig. 3g) and longitudinally at four (Fig. 3h) and eight weeks (Fig. 3i) after alpha particle therapy. Nonivasive monitoring and

analysis of disease reduction vs. untreated controls in a GEM model reinforce the therapeutic potential of this internalizing alpha particle emitting antibody construct in vivo.

**Alpha particles promote lethal feed-forward effect.** The kinetic profile of [$^{225}$Ac]hu11B6 accumulation in LNCaP-AR tumors is shown in the curve in Fig. 4a and highlights the increasing tumor uptake of drug with time (slope is 0.037 ± 0.001 %IA/g/h and intercept is 11.1 ± 0.25 %IA/g; $R^2$ is 0.999). Therapeutic efficacy of [$^{225}$Ac]hu11B6 is evidenced in the tumor volume data shown in Fig. 4b that compares drug vs. vehicle-treated control. After 5 days, the trajectory of untreated tumor volumes increases while tumors irradiated with alpha particles delivered by the hu11B6 antibody significantly decrease in volume. These data correlate with the survival curve shown in Fig. 1c in which half the animals in the treatment arm had no tumor at 120 days.

This response is due to the interaction of α-particles with tumor tissue that produces lethal DNA damage. The transcriptional response of the prostate cancer to this high LET damage may have a direct impact on the AR signaling axis and upregulate receptor expression. To investigate this potential synergy, AR upregulation in response to [$^{225}$Ac]hu11B6 therapy was

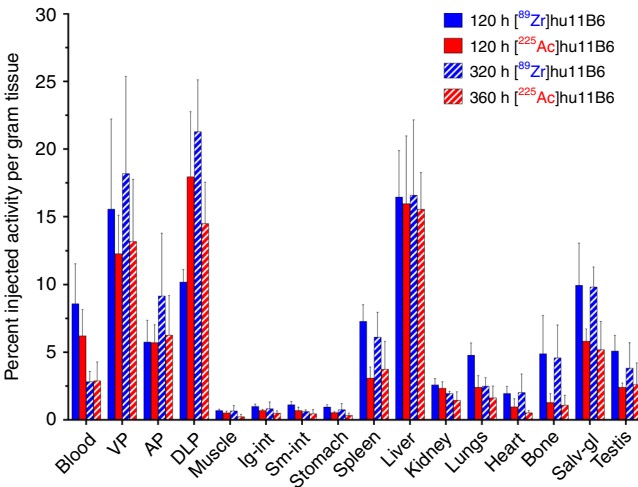

**Fig. 5** [$^{89}$Zr]hu11B6 is a surrogate PET reporter for [$^{225}$Ac]hu11B6 distribution in vivo. A pharmacokinetic comparison of [$^{89}$Zr]hu11B6 and [$^{225}$Ac]hu11B6 distribution in Hi-*Myc* x pb_*KLK2* mice shows that tissue uptake and blood clearance of the $^{89}$Zr-labeled diagnostic PET hu11B6 agent is comparable to the alpha emitting $^{225}$Ac-labeled therapeutic analog at both time points investigated. This analogous pharmacokinetic profile suggests that [$^{89}$Zr]hu11B6-PET can be utilized as a reporter for [$^{225}$Ac] hu11B6 therapy in vivo. The tumor accumulation of both agents in all three prostate lobes is similar

quantified in this animal model using Bioluminescent Imaging (BLI). The data in Fig. 4c show measured AR-driven luciferase bioluminescence signal (photons/s/mm$^2$/sr) vs. time in the treated and control animals. Interestingly, as shown in Fig. 4b, alpha irradiated tumor volume decreases with time while BLI signal increases (Fig. 4c). These data show upregulation of *AR* expression as a consequence of alpha irradiation with [$^{225}$Ac] hu11B6. Further, *AR* upregulation subsequently drives the overexpression of hK2 (*KLK2*) and PSA (*KLK3*) as shown in Fig. 4d. The subsequent overexpression of hK2 as a response to therapy creates a feed-forward loop that promotes binding more [$^{225}$Ac]hu11B6 as shown in the upward trajectory of tumor accumulation of drug (Fig. 4a) despite decreasing volume (Fig. 4b). The genotoxic insult from [$^{225}$Ac]hu11B6 irradiation initiates DNA-repair mechanisms in surviving cells, which further upregulates AR pathway activity.

**Pharmacokinetic similarity of $^{225}$Ac- and $^{89}$Zr-labeled hu11B6**. A comparison of the biodistribution at 120 and 360 h post injection of therapeutic [$^{225}$Ac]hu11B6 and its cognate reporter [$^{89}$Zr]hu11B6 was undertaken in the Hi-*Myc* x pb_*KLK2* mouse model (Fig. 5). These data clearly demonstrate that tissue uptake and blood clearance of the $^{89}$Zr-labeled diagnostic PET hu11B6 agent is comparable to the alpha particle emitting $^{225}$Ac-labeled therapeutic analog at both time points investigated. These analogous pharmacokinetic profiles verify that [$^{89}$Zr]hu11B6-PET can be used to estimate the dosimetric parameters for [$^{225}$Ac] hu11B6 therapy in man. Importantly, blood clearance, tumor accumulation in all three prostate lobes, and off-target sites are remarkably similar. Clearly the hu11B6 protein dominates the distribution of the tracer in vivo and suggests that both radio-metals are strongly bound by their respective chelates.

**Target specificity in nonhuman primates**. We evaluated the biodistribution of [$^{89}$Zr]hu11B6 in nonhuman primates based on the strong similarity of the pharmacokinetic profiles of $^{89}$Zr- and

$^{225}$Ac-labeled hu11B6 in rodents. Quantitative PET/CT imaging of the distribution of hu11B6 was performed in two adult male cynomolgus monkeys (8.3–9 y.o.; 10 kg) over the course of 16 days following a single IV administration of [$^{89}$Zr]hu11B6 (Fig. 6). The tracer cleared from the blood with an effective half-life of 2.5 days and accumulated in the prostate tissue of these animals, as expected from the rodent studies. Quantification of the noninvasive imaging was made by computing the standardized uptake value of [$^{89}$Zr]hu11B6 activity distribution in organs of interest. The uptake of tracer in prostate peaked 5–8 days post-administration with the ratio of prostate SUV to muscle SUV reaching 23-fold on day 8.

**Discussion**

Alpha particle drugs are now recognized as mainstream pharmacologic agents, delivering extremely large doses of ionizing radiation over microscopic dimensions corresponding to only a few cell diameters. FDA approval of Xofigo$^{TM}$ ([$^{223}$Ra]RaCl$_2$)[13], safe and effective pre- and clinical application of [$^{213}$Bi]Lintuzumab and [$^{225}$Ac]Lintuzumab[1, 3, 14], and compassionate use of [$^{225}$Ac]PSMA-617[15] have all demonstrated potent bioactivity with an ability to provide tumor control in man. When an alpha particle emitting radionuclide accumulates at the sites of malignancy, they locally deposit a high absorbed dose to the tumor target cells; normal healthy tissue is largely spared unless the delivery vehicle accumulates there in addition to tumor. The untoward off-target salivary and kidney uptake of [$^{225}$Ac]PSMA-617 may well limit the utility of this small molecule in treating prostate cancer as these healthy tissues also express PSMA[16]. This calls attention to the need for engineered biologic delivery molecules to truly disease tissue-specific targets.

The interaction of alpha particles with cells significantly perturbs numerous biological pathways thus altering homeostatic cell function. The hormone-DNA repair circuit is engaged in direct response to alpha-induced DNA damage and the intricate pathway networks that sense genomic damage become activated in order to initiate repair mechanisms and maintain genomic integrity[8, 9, 17]. Until now alpha particle drug discovery efforts have focused solely on inherent cytotoxicity and neglected the subtle, yet profound radiobiological changes that accompany irradiation. We have factored the mechanistic advantages of AR pathway upregulation into our drug design in order to enhance alpha therapy.

The androgen receptor pathway has an integral role in prostate cancer biology and manipulating AR expression is a viable strategy to eradicate this disease. Our data support a unique feed-forward mechanism that exploits the role of AR in prostate cancer resistance to therapy to become lethally addicted to [$^{225}$Ac] hu11B6 (Fig. 7). Herein, we demonstrate that cell specific alpha particle radiation has the potential to not only kill tumor cells, but also to induce DNA damage and upregulate *AR* and *KLK2*. We have constructed a system wherein [$^{225}$Ac]hu11B6 initiates a potent feed-forward loop that ultimately addicts the cancer to the lethal alpha particle emitting drug. Alternative strategies using small molecule AR inhibitors have become an important treatment option for late stage and castration-resistant prostate cancer[18]. Unfortunately, the bioactivity of these AR-inhibitors eventually fails as the tumor escapes through heterogenous mechanisms of acquired resistance[19].

The [$^{225}$Ac]hu11B6 drug is readily synthesized by the attachment of DOTA-chelated $^{225}$Ac and purified in a procedure developed to prepare clinical doses of [$^{225}$Ac]Lintuzumab[20]. Our SPR binding studies demonstrated clearly that the attachment of either chelate and radionuclide combination did not affect the $K_D$ of product relative to the native hu11B6. A [$^{89}$Zr]hu11B6 analog

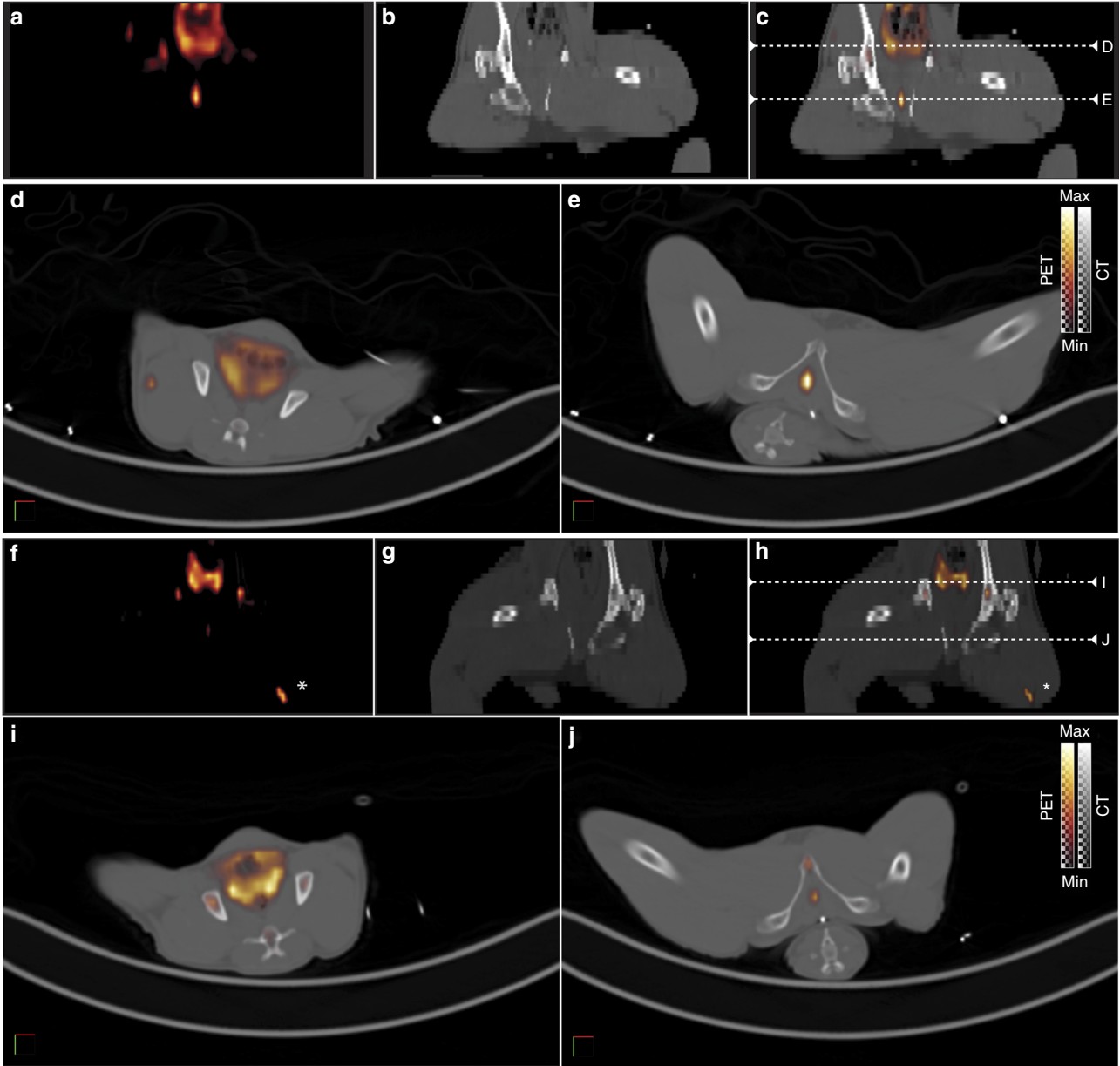

**Fig. 6** hK2-target specificity is imaged in nonhuman primate using [$^{89}$Zr]hu11B6 PET/CT. Prostate localization of diagnostic anti-hK2 tracer, [$^{89}$Zr]hu11B6, imaged in a representative healthy adult male cynomolgous monkey (9 y.o.; 9.9 kg) at 8 d (**a–e**) and 16 d (**f–j**) post IV administration. (**a–c, f–h**) PET, CT and fusion coronal images showing the location of two axial planes (white dashed lines) for corresponding axial plane images of the prostate (**d**, **i**) and epididymis (**e**, **j**). Nonspecific uptake at 16 d in left thigh (**f**, **h**) appears to localize to site of anesthetic administration (white asterisk)

provides robust correlative pharmacokinetic information that is used to report the behavior of the alpha emitting version of this drug in vivo. Two human xenograft models of prostate cancer (LNCaP-AR and VCaP) are used to demonstrate targeting specificity and efficacy. VCaP expresses more hK2 than LNCaP-AR and demonstrates greater accumulation of drug as predicted. The transgenic cancer susceptible mouse with prostate specific expression of human hK2 (Hi-*Myc* x pb_*KLK2*) recapitulates the biology of human prostate cancer and was engineered to investigate the radiobiological mechanisms that are the design foundation for [$^{225}$Ac]hu11B6. This GEM model shows higher *KLK2* expression in the ventral and dorsal lateral prostate lobes than the anterior, which correlates to the commonly observed rate of spontaneous adenocarcinoma development and intraepithelial neoplasia in the respective lobes of a Hi-*Myc* GEM. Our data

(Fig. 3a) confirm this hK2 tissue distribution and neoplastic behavior as evidenced by drug accumulation and is confirmed by MRI (Fig. 3f-h). The alpha emitting drug accumulates in the cancer present in these lobes and not in the nearby ureter and seminal vesicles (Fig. 3c) thus sparing these tissues due to the short alpha particle range. Specificity and efficacy of our hu11B6-mediated alpha therapy is apparent from the distinct lobe volume reduction when compared to control as a function of time (Fig. 3d-e).

The binding and internalization of hu11B6 has been engineered to perform several important tasks. First, targeting and binding hK2, an epitope expressed exclusively on prostate tissue and cancer in vivo; second, internalizing and transporting radionuclide cargo inside the cell to optimize the geometry of parent and progeny alpha decay. The hu11B6 binds to active,

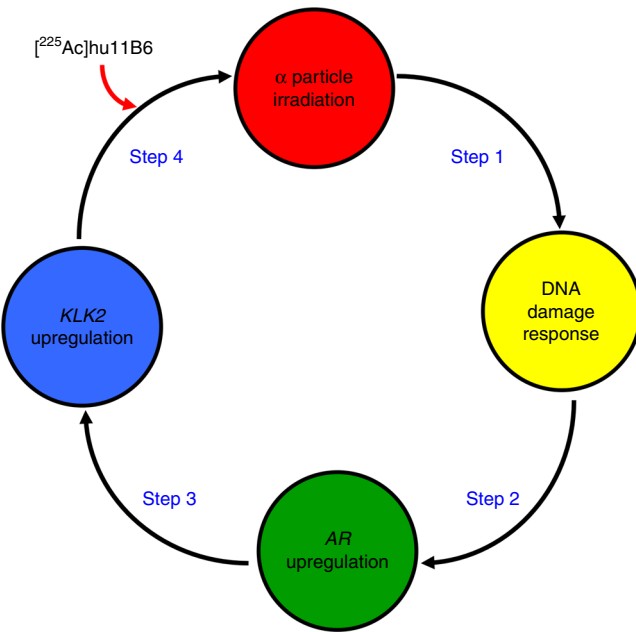

**Fig. 7** The mechanism of action in alpha particle-promoted feed-forward oncoaddiction. The interaction of α-particles with tumor tissue produces DNA damage. The ensuing transcriptional response prompts the hormone-DNA circuit to upregulate *AR* expression. *KLK2* expression subsequently upregulates due to *AR* and produces more hk2 target epitope for [$^{225}$Ac] hu11B6

membrane-bound hK2 and the ensuing immune complex is internalized by the cell and trafficked to lysosomal compartments in a process made possible by concomitant binding of the Fc fragment of hu11B6 with neonatal FcRn[6]. (n.b., hu11B6 does not bind to secreted hK2.) Loading an $^{225}$Ac payload onto hu11B6 produces an alpha nanogenerator drug with progeny that can yield 4 net alpha emissions[4, 14]. Cellular internalization of the hK2:[$^{225}$Ac]hu11B6 immune complex significantly improves the probability that the emitted alpha particles (of parent and progeny) will deposit energy inside the cancer cell. Indeed, in small animal models of osseous disease of bone-tropic prostate cancer, we are able to visualize the distribution of the [$^{225}$Ac]hu11B6 conjugate throughout the lesion (Fig. 2a). This is in contrast to the labeling of active bone surfaces, some of which are apposite the bone lesion, with the calcium mimetic $^{223}$Ra (Fig. 2b). As noted above, the short path length of alpha particles limits the radiobiological effect to only several cell diameters, however with high biological effectiveness due to high LET.

Molecularly specific radiotherapy through the antibody-conjugate approach in combination with internalization ensures that the alpha particle cascade inflicts significant DNA damage and prompts increased *AR* expression. This hormone-DNA circuit increases hK2 expression which establishes a feed-forward process leading the cancer to bind more [$^{225}$Ac] hu11B6. This mechanism is evidenced in the increasing tumor accumulation of [$^{225}$Ac]hu11B6 with time (Fig. 4a) despite the reduction in tumor burden (Fig. 4b). The predicted upregulation of *AR* is quantified in these tumors using the BLI readout that correlates with *AR* expression (Figs. 4c,d). The downstream effects of increased *AR* on *KLK2* and *KLK3* were measured (Fig. 4d). *FOLH1* which controls PSMA expression is notably not upregulated in these samples (Fig. 4d) and suggests that alpha particle therapeutics targeting PSMA expressed on prostate cancer (e.g., J591 or PSMA-617) will not engage a similar feed-forward mechanism.

Imaging the Ac-225 radionuclide has not been possible at the activities projected for clinical use. [$^{89}$Zr]hu11B6 is a positron emitting construct that we have evaluated as a surrogate reporter for [$^{225}$Ac]hu11B6. The biodistribution data obtained by tissue harvest and counted at $^{225}$Ac secular equilibrium is shown in Fig. 5. This study clearly shows that molecular hu11B6 directs the pharmacokinetic profile as evidenced by the strong parallel in $^{89}$Zr and $^{225}$Ac tissue and blood distribution and clearance values. The DFO and DOTA chelates stably contain $^{89}$Zr and $^{225}$Ac, respectively and are integral to construct design. With this validated noninvasive surrogate reporter platform for [$^{225}$Ac]hu11B6 in hand, we proceeded to investigate pharmacokinetics and safety in a nonhuman primate. The crab-eating macaque (*Macaca fascicularis*) also expresses hK2 in the prostate, albeit at 1000-fold lower levels than humans[21, 22], and it cross-reacts with the hu11B6 antibody. Our quantitative [$^{89}$Zr]hu11B6 PET data in adult male subjects (Fig. 6) showed that the tracer cleared from the blood with an effective half-life of 2.5 days and accumulated in their prostate tissue. Tracer uptake in prostate peaked 5–8 days post-administration and on day 8 the prostate-to-muscle SUV ratio was 23. This study established the hK2-expressing prostate targeting ability of hu11B6 antibody in a healthy nonhuman primate model. Accumulation of [$^{89}$Zr]hu11B6 even at this extremely low-level of hK2 in the macaque indicates that robust targeting in humans of the therapeutic construct should have the opportunity to provide significant therapeutic benefit.

Our overall strategy has been designed to take full advantage of the unique consequences of [$^{225}$Ac]hu11B6 tissue-specific targeting and high LET α-particle irradiation of prostate cancer. We recognized that the alpha particle induced DNA damage and relied on ensuing upregulation of AR and *KLK2* to addict the prostate cancer to the therapeutic [$^{225}$Ac]hu11B6 agent. Radio-biological addiction favorably drives the cancer to bind more of the cytotoxic drug, resulting in ablation of the disease.

## Methods
**Cell lines**. The VCaP cell line was purchased from American Type Culture Collection (Manassas, VA) and cultured according to the manufacturer's instructions. LNCaP-AR-luc (LNCaP cell line with over-expression of wild-type AR expressing luciferase under the control of ARR2-Pb) was a kind gift from the laboratory of Dr. Charles Sawyers, which previously developed and reported the cell line[23]. These lines are routinely surveyed for contamination by scrutinizing morphology, cell kinetics and testing for mycoplasma.

**Transgenic *KLK2* mouse models**. A transgenic mouse model expressing prostate tissue specific activated hK2 was used. Here, site-directed mutagenesis of APLILSR to APLRTKR at positions 4, −3, and −2 the zymogen sequence of *KLK2* was performed using a Quick Change Lightning Mutagenesis Kit (Stratagene). This enabled furin, a ubiquitously expressed protease in rodent prostate tissue, to efficiently cleave the short activation peptide at the cleavage site (−1 Arg/ + 1 Ile), resulting in enzymatically active hK2. Sequencing was performed to verify the genotype using the following primers: 5′-TTC TCT AGG CGC CGG AAT TA-3′ (forward), 3′-CCC GGT AGA ATT CGT TAA CCT-3′ (reverse). This construct was cloned into a SV40 T-antigen cassette downstream of the short rat probasin (pb) promoter and microinjected into fertilized mouse embryos (C57BL/6) and implanted into pseudopregnant female mice. A cancer-susceptible transgenic mouse model with prostate specific hK2 expression was created by crossing the pb_KLK2 transgenic model with the Hi-*Myc* model (ARR2PB-Flag-Myc-PAI transgene). The Hi-*Myc* x pb_*KLK2* GEM model used in the studies has previously been described in detail[10]. Integration of genes into the genome of the offspring was confirmed by Southern blot analysis and PCR. Mice were monitored closely in accordance with IACUC-established guidelines and RARC animal protocol (# 04-01-002).

**Animal studies**. All animal experiments were conducted in compliance with institutional Animal Care and Use Committee (IACUC)-established guidelines at Memorial Sloan Kettering Cancer Center (MSKCC). For xenograft studies: male athymic BALB/c nude mice (6–8 weeks old, 20–25 g) were obtained from Charles River. VCaP and LNCaP-AR-luc tumors were inoculated in the right and left flanks by subcutaneous injection of $1-5 \times 10^6$ cells in a 200 μL cell suspension of a 1:1 v/v mixture of media with Matrigel (Collaborative Biomedical Products, Inc.). Tumors

developed after 3–7 weeks. Osseous tumors were established in the tibia of the right hindlimb of BALB/c nude mice using LNCaP-AR, as previously reported[24], and monitored by bioluminescent imaging.

**Pharmacokinetic tissue distribution.** Biodistribution studies were conducted to evaluate the uptake and pharmacokinetic distribution of Zirconium-89 labeled hu11B6 ([$^{89}$Zr]hu11B6) in human prostate cancer xenograft and GEM models. Mice received a single 5.55 MBq dose of [$^{89}$Zr]hu11B6 (0.150 mCi of activity on 5–50 µg of protein) or a single 11.1 kBq dose of [$^{225}$Ac]hu11B6 (300 nCi on 5 µg antibody) for injection via intravenous tail-vein injection ($t = 0$ h). Animals ($n = 4$–5 per group) were euthanized by CO$_2$ asphyxiation at 4, 48, 120 and 360 h post-injection of [$^{225}$Ac]hu11B6; or at 120 and 340 h post-injection of reporter [$^{89}$Zr]hu11B6. Blood was immediately harvested by cardiac puncture. Tissues (including the tumor) were removed, rinsed in water, dried on paper, weighed, and counted on a gamma-counter using a 370–510 KeV window at secular equilibrium. Aliquots (0.020 mL) of the injected activities were used as decay correction standards and background signal was subtracted from each sample. The percentage of injected activit per gram of tissue weight (%IA/g) was calculated for each animal and data plotted as mean ± SD. Statistical analysis of data was performed using Prism software (Graphpad Software Inc, La Jolla, CA).

**Antibodies.** Humanized 11B6 (hu11B6) and recombinant mutant- hu11B6H435A (modified at Histidine 435 to Alanine; H435A-hu11B6) to abrogate FcRn binding[25] was developed by DiaProst AB (Lund, Sweden). hu11B6 and hu11B6H435A used for Actinium-225 labeling was produced by Innovagen AB (Lund, Sweden). The DFO conjugated hu11B6 construct used for $^{89}$Zr-labeling was produced by Fujifilm Diosynth Biotechnologies UK Ltd (Billingham, United Kingdom).

**Bioluminescence imaging.** In vivo AR-activity was recorded by measuring luciferase activity of LNCaP-AR/luc s.c. bilateral xenografts in BALB/c nude mice. These tumors co-expressed exogenous AR and the AR-dependent luciferase reporter vector ARR2-Pb-Luc. Mice were placed under anesthesia with isoflurane prior to retro-orbital injection of 10 µL D-Luciferin (30 mg/mL, dissolved in PBS). A sequence of images with a range of acquisition times was acquired immediately after injection. Radiance (photons/s) was recorded (using Living Image® 4.5.2) from each individual tumor and divided by its respective volume measured by caliper (mm$^3$).

**Therapy studies.** Study I examined a single 300 nCi activity of [$^{225}$Ac]hu11B6, [$^{225}$Ac]hu11B6H435A or $^{225}$Ac-labelled non-specific huIgG$_1$ injected in three groups of ten male LNCaP-AR s.c. mice. Length (l) and width (w) of the tumors were measured by caliper and the volume for a rotated ellipsoid ($V = \frac{1}{2}$ w2l) was calculated. Weight loss of 20% or a tumor diameter exceeding 15 mm was set as an endpoint. Study II examined two groups of 15–16 week old male Hi-*Myc* x pb_*KLK2* GEM that were randomly selected for specific treatment with 300 nCi of [$^{225}$Ac]hu11B6 ($n = 6$) or vehicle ($n = 6$) at week 40. Tumor progression in each animal in both treatment arms was followed by longitudinal MRI. Information on treatment was blinded to the readers analyzing the imaging data. Volumetric MRI measurement of the three prostate lobes was collected over time and compared to tumor growth progression between [$^{225}$Ac]hu11B6 and non-treated Hi-*Myc* x pb_*KLK2* animals.

**Competitive binding of [$^{225}$Ac]hu11B6 and [$^{89}$Zr]hu11B6.** Biotinylated 11B6 (100 µL; 2 mg/L) was added to streptavidin-coated microtiter plates, followed by 1 h of incubation with shaking. The plate was washed, after which 20, 100, 200, 400 or 1000 µg of compound (antibody) in 100 µL of DELFIA Assay Buffer was added to the wells, in duplicates, to compete with the capture antibody. Samples containing 0.34 ng/mL or 3.4 ng/mL in 100 µL of DELFIA Assay Buffer were then added to the wells. After 2 h incubation with shaking, the plate was washed, and the Eu$^{3+}$ labeled tracer antibody 6H10 was added (200 µL; 0.5 mg/L). The plate was incubated for 1 h with shaking and then washed. DELFIA Enhancement Solution (200 µL) was added and the time-resolved fluorescence was measured 5 min later.

**RNA isolation and quantitative PCR.** Tissue samples were flash-frozen and immediately used for RNA extraction. Approximately 20 mg of tumor sample was used for RNA extraction. Samples were homogenized using a bullet blender in 600 µL of buffer RLT (RNeasy mini kit) with β-mercaptoethanol. The lysates were centrifuged for 3 min at full speed and the supernatant was used for RNA extraction using RNeasy mini kit (QIAGEN). On column DNA digestion was done using RNase-Free DNase set (QIAGEN). RNA quality and quantity was determined using a spectrophotometer at 260 and 280 nm (Nanodrop-2000, Thermo Scientific). cDNA was generated using the High Capacity cDNA Reverse Transcription Kit (Applied Biosystems; Life Technologies). Quantitative-PCR was done using RT2 SYBR Green Fluor qPCR Mastermix and RT2 qPCR primers (Qiagen) on a CFX96 Touch Real-Time PCR Detection System (Bio Rad). *KLK2* (hK2), *KLK3* (PSA), *AR* (androgen receptor), *FOLH1* (PSMA)

expression was quantified relative to *ACTB* (beta actin) using the comparative CT method.

**Tissue histology and autoradiography.** After mice were euthanized a tissue package containing prostate lobes, seminal vesicles and prostatic urethra was surgically excised and incubated in Tissue-Tek optimal cutting temperature compound (Sakura Finetek USA, Inc) on ice for 45 min, and then snap-frozen on dry ice in a cryomold. Sets of contiguous 15 µm thick tissue sections were cut using a CM1950 cryostat microtome (Leica Microsystems Inc) and arrayed onto Super-frostPlus glass microscope slides (Thermo Scientific). The slides were exposed for 144 h on phosphor plates and read by a Fujifilm BAS-1800II bio-imaging analyzer (Fuji Photo Film Co.) generating digital images with 50 µm pixel dimensions. Digital images were obtained with an Olympus BX60 System Microscope (Olympus America, Inc.) equipped with a motorized stage (Prior Scientific, Inc.).

**Magnetic resonance imaging.** We used MRI to assess treatment response in our prostate GEM models. Mouse prostate MR images were acquired on a Bruker 4.7 T Biospec scanner operating at 200 MHz and equipped with a 400 mT/m ID 12 cm gradient coil (Bruker Biospin MRI GmbH, Ettlingen, Germany). A custom-built quadrature birdcage resonator with ID of 32 mm was used for RF excitation and acquisition (Stark Contrast MRI Coils Research Inc., Erlangen, Germany). Mice were anesthetized with oxygen and 1% isoflurane gas. Animal breathing was monitored using a small animal physiological monitoring system (SA Instruments, Inc., Stony Brook, New York). T2 weighted scout images along three orthogonal orientations were first acquired for animal positioning. The T2-weighted fast spin-echo RARE sequence (rapid acquisition with relaxation enhancement) was used to acquire axial mouse pelvic images with a slice thickness of 0.8 mm, FOV 40 mm × 35 mm with a spatial resolution of 117 mm × 133 mm using the following acquisition parameters: TR = 3824.4 s, TE = 48 ms, RARE factor 8 for an average of eight scans. Following on-line reconstruction, data were exported to FIJI (NIH) the area of each individual lobe was assessed. Cystic dilations were not included in our volumetric measurements.

**Positron emission tomography.** PET/CT imaging of male crab-eating macaque (Charles River) was performed with a Siemens Biograph64 mCT PET/CT system. PET images were acquired at 1, 24, 72, 144, 216, and 384 h after intravenous injection (IV) of 10 mg of 2.50 mCi (92.5 MBq) [$^{89}$Zr]hu11B6. Whole body acquisitions consisting of six bed positions covering from the crown of the animal's head to mid-tibia. The head and feet were scanned with 4 min per bed, while the beds over the pelvic region were scanned with 14 min per bed, for a total imaging time (including gaps between scans) of about 44 min. Animals were maintained under 2% isoflurane/oxygen anesthesia during the scanning. Data was exported in raw format and the rigid body (3 degrees of freedom) co-registration between PET and CT data was performed in Amira 5.3.3 (FEI). Amira and FIJI were used to produce the majority of the figures in the manuscript. Siemens Syngo Multi-Modality Workplace software (version VE40A) was used to analyze and quantify [$^{89}$Zr]hu11B6 uptake on PET images.

**[$^{223}$Ra]RaCl$_2$ and [$^{225}$Ac]hu11B6 bone metastasis localization.** Using an intratibial inoculation model of metastasis, we evaluated the acute microdistribution of [$^{225}$Ac]hu11B6 compared to [$^{223}$Ra]RaCl$_2$ in intratibial lesions of LNCaP-AR, which is a mixed osteolytic and osteoblastic bone lesion that we have previously evaluated and reported[10, 24]. Tumor growth was monitored by bioluminescence and x-ray computed tomography (CT) using the IVIS SpectrumCT (Perkin Elmer; Waltham, MA). The tumor bearing hindlimb of the mice were cryosectioned 24 and 120 h following injection by [$^{223}$Ra]RaCl$_2$ (150 nCi; 5.5 kBq) and [$^{225}$Ac]hu11B6 (300 nCi; 11.1 kBq), respectively, and scanned by α-camera, as previously described[26]. Briefly, no fixation or decalcification chemicals were applied to mouse tissues, which were either stored (−80º C) or flash-frozen in liquid nitrogen. Autoradiography was performed on fresh-frozen sections cut on a custom-modified (cryo-cooled) Leica 1860 cryostat at 10 µm. Alpha camera images consist of undecalcified tissue sections that were placed in direct contact with silver activated zinc sulfide scintillant (Eljen Technologies) and imaged using a cryo-cooled EMCCD (Photometrics CascadeII). The adjacent section was then processed for Safranin-O development for bone mineral and proteoglycan localization.

**Radiochemistry.** $^{225}$Ac (ORNL, Oak Ridge, TN) was conjugated to the hu11B6 antibody or isotype-matched control antibody and purified using a 2-step labeling procedure[4, 20]. Activity was measured at secular equilibrium with a Squibb CRC-17 Radioisotope Calibrator (E.R. Squibb and Sons, Inc., Princeton, NJ) set at 775 and multiplying the displayed activity value by 5. The radiochemical purity of the final product, [$^{225}$Ac]hu11B6, was determined using instant thin-layer chromatography (ITLC) with a stationary phase of silica gel impregnated paper (Gelman Science Inc., Ann Arbor, MI) and two different mobile phases. Mobile phase I is 10 mM ethylenediaminetetraacetic acid and II is 9% sodium chloride/10 mM sodium hydroxide. The strips were counted in a Packard Cobra γ-counter (Packard Instrument Co., Inc., Meriden, CT) using a 370–510 KeV window. The purified radioimmunoconstruct was formulated in a solution of 1% human serum albumin (HSA, Swiss Red Cross, Bern, Switzerland) and 0.9% sodium chloride (Normal

Saline Solution, Abbott Laboratories, North Chicago, IL) for intravenous injection. Radiochemically pure $^{223}$Ra was eluted from an Actinium-227 source, as described[27] and formulated in 0.03 M citrate in saline. An empirically derived calibrator setting of #277, using a CRC-127R dose calibrator (Capintec Inc), was used to dose $^{223}$Ra.

$^{89}$Zr (MSKCC Radiochemistry and Imaging Probes (RMIP) Core Facility or 3D Imaging, Little Rock AR) was conjugated to a hu11B6-DFO antibody as previously described[10, 24]. Activity was measured with a Squibb CRC-17 Radioisotope Calibrator set at #465. Briefly, 8.5 mCi of $^{89}$Zr[Zr]oxalate was neutralized with 0.025 mL of 1 M Na$_2$CO$_3$ to pH 7.0. Any colloidal precipitate was removed with centrifugal filtration (0.22 μm Corning Spin-X filter at 1000×g) and 0.25 mL of 11B6-DFO (8.8 g/L; Fuji Film; lot # NBS0131-6-2 (Prost B3298) in 25 mM sodium acetate buffer (pH 5.5) was added to the clear neutralized $^{89}$Zr[Zr] filtrate. The reaction mixture was purified after 75 min at ambient temperature by size exclusion chromatography using a 10DG column (BioRad) mobile phase and a 1% HSA mobile phase. The radiochemical purity of the final product, [$^{89}$Zr]hu11B6, was assayed using ITLC (silica gel impregnated paper and Mobile Phases I and II) as described above.

**Binding affinity of labeled and unlabeled hu11B6.** The binding kinetics for the association ($k_a$), dissociation ($k_d$) and affinity constant ($K_D$) of native hu11B6 vs. both [$^{225}$Ac]hu11B6 and [$^{89}$Zr]hu11B6 radioimmunoconjugates were determined using surface plasmon resonance (SPR) on a Biacore 2000 (GE Healthcare, Marlborough, MA). Separate SPR experiments were performed wherein the unmodified hu11B6 antibody, the intermediate immunoconjugates (DOTA-hu11B6 or DFO-hu11B6) and the final [$^{225}$Ac]hu11B6 and [$^{89}$Zr]hu11B6 radioimmunoconjugates were used as ligands captured on a Protein A sensor chip (29127558, GE Healthcare). Antibody capture was accomplished by diluting the unmodified hu11B6 antibody and the various immunoconjugates to a concentration of 1 μg/mL in HBS-EP buffer (BR100188, GE Healthcare), and flowing the solution over a protein A sensor chip for 1 min at a flow rate of 5 μL/min. Recombinant human hK2 (Turku University, Finland)[28] was used as the analyte that was flowed over the protein A sensor chip having the captured hu11B6 antibody or derivative. The binding kinetics of the antibody (ligand) captured on the sensor chip was evaluated across a concentration series of the hK2 antigen (analyte) at 0, 3.125, 6.25, 12.5, 25, 50, 100, and 200 nM in HBS-EP buffer. Each concentration of analyte was injected for 5 min at a flow rate of 5 μL/min to allow it to bind to the antibody captured on the protein A sensor chip. Next, the binding buffer (HBS-EP) was allowed to flow over the sensor chip for 15 min (5 μL/min) to allow dissociation of the antigen from the 11B6 antibody-hK2 antigen immunocomplex on the chip. Finally, regeneration buffer (10 mM glycine/HCl, pH 2.5) was passed over the chip surface for 1 min (5 μL/min) to effect complete dissociation of captured antibody. HBS-EP buffer was flown (5 μL/min) over the chip for 2 min to stabilize the protein A chip surface prior to the injection of the next sample in the concentration series of the analyte as described above. Biacore control software 3.2 was used to analyze the kinetic data and the 1:1 binding with mass transfer fit was used to derive kinetic constants for the interaction between the various 11B6 immunoconjugates and purified hK2 antigen.

**Statistical analysis.** Statistical analysis of data was performed using Prism 6 software (Graphpad Software Inc, La Jolla, CA). Data are displayed as means with their standard errors (SEM). P-values for comparisons between treatment groups were obtained in GraphPad Prism using Student's t-test for statistical significance (unpaired, two-tailed t-test). p-Values <0.05 were considered statistically significant. Log-rank (Mantel–Cox) and Mantel–Haenszel tests were used in Prism software to compare survival outcomes.

**Data availability.** All data generated or analyzed during this study are available within the article and Supplementary Files, or available from the authors upon request.

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

## Acknowledgements

The authors thank Dr. David A. Scheinberg for his intellectual input. For technical support and expertise we thank Dr. Pat Zanzonico, director and Dr. Mihaela E. Lupu, of the MSKCC Small Animal Imaging Core Facility, Mr. Muc Du and Mr. Simon Marim of the Cornell Bioimaging Core Facility, and Dr. Heather Martin of the MSKCC veterinary staff. Isotopes used in this research were supplied in part by the United States Department of Energy Office of Science, Isotope Program, Office of Nuclear Physics. This study was supported in part by the Imaging and Radiation Sciences Program (IMRAS), US National Institutes of Health (NIH) grants P30 CA008748 (MSK Cancer Center Support Grant) and P30 CA006973 (Johns Hopkins University Cancer Center Support Grant). The MSKCC Small-Animal Imaging Core Facility is supported in part by NIH grants P30 CA008748-48, S10 RR020892-01, S10 RR028889-01 and the Geoffrey Beene Cancer Research Center. We

also acknowledge Mr. William H. Goodwin and Mrs. Alice Goodwin and the Commonwealth Foundation for Cancer Research, the Experimental Therapeutics Center and the Radiochemistry & Molecular Imaging Probe Core (P50-CA086438), all of MSKCC. M.R.M. was supported by NIH R01CA166078, R01CA55349, P30CA008748, P01CA33049, F31CA167863, the Memorial Sloan Kettering Center for Molecular Imaging and Nanotechnology (CMINT), Mr. William H. and Mrs. Alice Goodwin and the Commonwealth Foundation for Cancer Research, and The Center for Experimental Therapeutics of Memorial Sloan Kettering Cancer Center. D.L.J.T. was supported by the Steve Wynn Prostate Cancer Foundation Young Investigator Award (PCF-YIA), the Patrick C. Walsh Fund, and NIH R01CA201035. D.U. was supported, in part, by the Knut and Alice Wallenberg Foundation, the Bertha Kamprad Foundation, and the David H. Koch PCF-YIA. S. M.L. was supported by the Ludwig Center for Cancer Immunotherapy at MSKCC and the National Cancer Institute (P50-CA86438). S.E.S. was supported by the Swedish Cancer Society and Swedish National Health Foundation (ALF) and Swedish Research Council. H. L. was supported, in part, by the National Cancer Institute [R01CA160816, R01 CA175491], the National Institute for Health Research (NIHR) Oxford Biomedical Research Centre Program in the UK, the Swedish Cancer Society (project no. 14-0722), and the Swedish Research Council (VR-MH project no. 2016-02974). P.T.S. and H.L. were supported in part by the MSKCC SPORE in Prostate Cancer (P50 CA92629), the David H. Koch Fund of the PCF, and the Sidney Kimmel Center for Prostate and Urologic Cancers.

## Author contributions

M.R.M., D.L.J.T. and D.U. conceived, designed, performed, and analyzed all the experiments and data. T.H., T.G., S.K.S. and T.M.K. performed the analyses, data acquisition and analysis; O.V.T. in study design. D.R.V. and D.S.A. assisted in study design and radioconjugate formulation. P.A.W. and B.J.B. assisted in technical study design and data analysis. S.E.S., J.S.L., P.T.S., H.S., H.G.L. and S.M.L. supervised the aspects of the project. All authors discussed the results, prepared, commented and approved the manuscript.

## Additional information

**Competing interests:** Authors D.L.J.T., S.E.S., H.L. and D.U. are shareholders of Diaprost Inc., which owns the antibody hu11B6. The remaining authors declare no competing interests.

