## [Peer Review File · Nature Communications]

Reviewers' comments:

Reviewer #1 (Remarks to the Author):

The authors are to be commended for a technically sophisticated series of experiments to qualify response to an actinium-KLK2 antibody conjugate. This has clearly involved a great investment of resources and time not least to engineer a transgenic model of prostate cancer overexpressing both Myc and KLK2. In order to capitalise fully on these studies the authors need to show that this therapeutic approach outperforms other AR-targeting strategies (eg. Enzalutamide or Apalutamide treatment) both of the models they have used here but also of models not engineered to have a KLK2 addiction. For example, Hi-Myc on its own or a PTEN/p53 knockout. Alternatively the authors need to show that their therapeutic has an impact on anti-androgen resistant derivatives of current cell-lines/models. Without this additional contextual information it is difficult to gauge the translational potential of these findings.

Reviewer #2 (Remarks to the Author):

In this paper, McDevitt and co-authors provide convincing evidence that humanized antibody against hK2 labeled with Actinium-225 can target prostate cancer cells in vivo resulting in reduction of tumor burden. This is very thorough study. The only comment I have is that the models used probably do not represent cancer in man in all aspect (this is the problem with models). The VCaP cells express very highly AR due to the AR gene amplification. The LNCaP cells used had been stable transfected with AR and the GEM was forced to express active hK2. In real life, the tumors and metastases probably contain cells that are expressing heterogeneous levels of AR and hK2, ie some cells express highly some lowly. However, this paper is an excellent basis for future work.

Reviewers' comments:

Reviewer #1 (Remarks to the Author):

The authors are to be commended a for a technically sophisticated series of experiments to qualify response to an actinium-KLK2 antibody conjugate. This has clearly been involved a great investment of resources and time not least to engineer a transgenic model of prostate cancer overexpressing both Myc and KLK2. In order to capitalise fully on these studies the authors need to show that this therapeutic approach outperforms other AR-targeting strategies (eg. Enzalutamide or Apalutamide treatment) both of the models they have used here but also of models not engineered to have a KLK2 addiction. For example, Hi-Myc on its own or a PTEN/p53 knockout. Alternatively the authors need to show that their therapeutic has an impact on anti-androgen resistant derivatives of current cell-lines/models. Without this additional contextual information it is difficult to gauge the translational potential of these findings.

We thank the reviewer for their positive assessment of our work, and the effort put forward to build models and acquire long term therapy data on this novel cancer treatment approach. Reviewer #1 also raises an important question, namely how well the actinium-225 labeled antibody conjugate targeting hK2 will work in cancers that have acquired resistance to standard anti-androgen treatment. We would envision that initial clinical trials using this agent would commence only in patients that have failed conventional treatments, similar to the clinical trial requirements for new targeted agents. Thus, ²²⁵Ac-hu11B6 would not be compared head to head with enzalutamide. However, we predict that the novel approach described here will still display efficacy against such aggressive cancers.

Our treatment strategy is dependent upon AR expression, but only in the sense that AR drives KLK2 expression, which provides the localization and uptake mechanism for the cytotoxic antibody. Importantly, our approach differs from conventional prostate cancer treatments that target AR directly. Significantly, even in heavily-treated patients with metastatic castration resistant prostate that acquire resistance to androgen deprivation therapeutic (ADT) agents, there is strong selective pressure for the continued expression of active AR. Indeed, 62.7% of patients in a recent study displayed genomic aberrations in AR, predominantly gene amplification but also gain-of-function ligand binding domain mutations (data from cbiportal.org - Robinson D, et al. Cell 2015, 161(5): 1215-1228). Importantly, nearly all of these cancers continue to express high KLK2 (display item, below).^{1,2}

Hence, AR activity with corresponding KLK2 expression is predominantly maintained in ADT resistant prostate cancer, so we predict that the use of our actinium-225 labeled antibody conjugate targeting hK2 will be applicable in this disease setting and its uptake and effect increased.

With respect to the development or use of additional models; an issue exists with the fact that rodents do not contain the KLK2 gene. Therefore, the mouse does not express prostate specific kallikreins (KLK2 is only expressed in humans, dogs and old world monkeys). Thus, evaluation of Myc or PTEN/p53^{+/-} on its own would not provide significant biological value or insight. It should be noted that the genetically engineered models used in this work express hK2 at levels *below* that observed in man. To confirm that the uptake of the radiopharmaceutical is not specific for this engineered system (or subcutaneous tumors), we evaluated tracer uptake of ⁸⁹Zr-hu11B6 in cynomolgus monkeys. These primates naturally express the hK2 target (that cross reacts with the hu11B6 antibody), again at levels significantly lower than that seen in man. Rather - the reviewer writes that the models that we have developed and used in this study have a "KLK2 addition", this is not the case, and any error in interpretation is the fault of the authors. We have revised the manuscript in several places to make clear the following: A radiopharmaceutical therapy targeting a product of the AR-axis (the hK2 protein) exploits a genomic feedback loop in prostate cancer cells which up regulate AR (and therefore the target) in response to DNA damage.

Reviewer #2 (Remarks to the Author):

In this paper, McDevitt and co-authors provide convincing evidence that humanized antibody against hK2 labeled with Actinium-225 can target prostate cancer cells in vivo resulting in reduction of tumor burden. This is very thorough study. The only comment I have is that the models used probably do not represent cancer in man in all aspect (this is the problem with models). The VCaP cells express very highly AR due to the AR gene amplification. The LNCaP cells used had been stable transfected with AR and the GEM was forced to express active hK2. In real life, the tumors and metastases probably contain cells that are expressing heterogeneous levels of AR and hK2, ie some cells express highly some lowly. However, this paper is an excellent basis for future work.

We again thank the reviewer for their kind comments regarding the manuscript. Indeed, humanized antibody 11B6 against hK2, labeled with actinium-225, can target prostate cancer cells to effectively reduce tumor burden in vivo in multiple models of the disease. We especially appreciate the comment regarding the 'thoroughness' of the work. The reviewer also makes a comment about the molecular heterogeneity of prostate cancer, an inherent fact for likely all cancers. This underlines several aspects of the approach that we have pursued - with translational intent. Utilizing a potent, multiple alpha particle emitting radionuclide conjugate, such as actinium-225, provides an intense bystander effect upon decay. Alpha particles (of energies between 2-4 MeV) will travel several tens of microns in water (or tissue) enabling significant impact to cells surrounding the site of uptake. This circumscribes the radiobiological effect to only neighbouring (likely cancer or tumor microenvironmental-supporting cells).

It also bears repeating that the fundamental response to genotoxic insult of prostate cancer cells to radiotherapeutic DNA damage. Namely, we have observed that following the application of alpha particle emitters targeted to the cancer cells, there is a significant upregulation of the androgen receptor and the targeted downstream protein (hK2). Thus, this positive feedback loop drives greater accumulation to the disease site for enhanced radiobiological effect. As well, please see the response to comments from Reviewer #1 (above) that detail findings from the AR copy number variation in patients that describe amplification and gain of function changes in populations that are resistant to conventional treatments. This should further enhance the capacity of the ²²⁵Ac-hu11B6 agent to achieve responses in late-stage prostate cancer patients in a Phase 0/I currently in the planning phase.

Reference list:

1. Cerami et al. The cBio Cancer Genomics Portal: An Open Platform for Exploring Multidimensional Cancer Genomics Data. *Cancer Discovery*. May 2012 2; 401
2. Gao et al. Integrative analysis of complex cancer genomics and clinical profiles using the cBioPortal. *Sci. Signal*. 6, p11 (2013)

REVIEWERS' COMMENTS:

Reviewer #1 (Remarks to the Author):

Hopefully this will be given the opportunity for a clinical trial.

Reviewer #2 (Remarks to the Author):

This is a submission of revised version of a manuscript. I did not have specific wishes for revision for the first submission. Authors have now also nicely responded to my concern in terms of tumor heterogeneity. Indeed the bystander effect is an important issue and may help to overcome the problem of heterogeneity.